# Application of Synephrine to Grape Increases Anthocyanin via Production of Hydrogen Peroxide, Not Phytohormones

**DOI:** 10.3390/ijms25115912

**Published:** 2024-05-29

**Authors:** Masaya Suzuki, Aoi Kimura, Shunji Suzuki, Shinichi Enoki

**Affiliations:** Laboratory of Fruit Genetic Engineering, The Institute of Enology and Viticulture, University of Yamanashi, 1-13-1 Kofu, Yamanashi 400-0005, Japan; g21lf006@yamanashi.ac.jp (M.S.); l16fsw08@yamanashi.ac.jp (A.K.)

**Keywords:** *CAT*, *Chit4*, global warming, grape skin coloration, high-quality grape, *mybA1*, sugar/acid ratio, *SOD3*, *UFGT*

## Abstract

Global warming has caused such problems as the poor coloration of grape skin and the decreased production of high-quality berries. We investigated the effect of synephrine (Syn) on anthocyanin accumulation. Anthocyanin accumulation in cultured grape cells treated with Syn at concentrations of 1 mM or higher showed no significant difference, indicating that the accumulation was concentration-independent. On the other hand, anthocyanin accumulation was dependent on the compound used for treatment. The sugar/acid ratio of the juice from berries treated with Syn did not differ from the control. The expression of anthocyanin-biosynthesis-related genes, but not phytohormones, was increased by the treatment with Syn at 24 h or later. The Syn treatment of cultured cells increased *SOD3* expression and hydrogen peroxide (H_2_O_2_) production from 3 to 24 h after treatment. Subsequently, the expression of *CAT* and *APX6* encoding H_2_O_2_-scavenging enzymes was also increased. Treatment of cultured cells with Syn and H_2_O_2_ increased the expression of the H_2_O_2_-responsive gene *Chit4* and the anthocyanin-biosynthesis-related genes *mybA1* and *UFGT* 4 days after the treatment and increased anthocyanin accumulation 7 days after the treatment. On the other hand, the treatment of berries with Syn and H_2_O_2_ increased anthocyanin accumulation after 9 days. These results suggest that Syn increases anthocyanin accumulation through H_2_O_2_ production without changing phytohormone biosynthesis. Syn is expected to improve grape skin coloration and contribute to high-quality berry production.

## 1. Introduction

Preventing the global-warming-induced decrease in crop quality is an urgent issue. Grapevine (*Vitis* spp.) is an economically important plant widely grown globally for wine production and consumption as table grapes. The increase in average temperature due to global warming has decreased grape skin coloration by inhibiting anthocyanin accumulation [1,2]. It is predicted that further increases in average temperature would cause significant economic damage not only to grape growers but also the winemaking industry [3]. Thus, it is necessary to prevent the decrease in grape skin coloration due to global warming.

Simple cultivation techniques for preventing grape skin coloration decrease are desired because the existing methods, such as girdling [4], leaf removal [5,6], and cluster thinning [7], require specific skills and intensive labor. The direct application of biologically active compounds that increase grape skin coloration, such as allantoin [8], amino acids [9], and vanillyl acetone [10], has stirred up interest in recent years. Therefore, we screened for biologically active compounds and found that synephrine (Syn) increases grape skin coloration (Enoki, personal communication). Syn (4-[1-hydroxy-2-(methylamino)ethyl]phenol) is an alkaloid with a phenethylamine skeleton. It is found in some orange species [11,12] and used as a dietary supplement because of its lipolytic effect [13,14]. However, there are no reports on its effects on crops. To develop new technology for increasing grape skin coloration during ripening, it is necessary to examine and clarify the mechanism of the coloration effect of Syn.

Such phytohormones as abscisic acid (ABA), ethylene (ET), and jasmonic acid (JA) increase berry skin coloration in response to sunlight [15], water [16], and temperature [17]. For example, ABA and ET promote ripening [8,10,18,19], which is characterized by anthocyanin accumulation and increased sugar content in the berry, whereas JA increases disease resistance but increases anthocyanin accumulation as a side effect [20]. ABA, but not ET, increases anthocyanin accumulation during grape berry ripening [4,21,22,23,24,25,26]. Therefore, we hypothesize that Syn increases grape skin coloration via ABA.

In this study, we clarified the mechanism of Syn-mediated skin coloration for grape quality improvement. We performed field studies on the coloration effects of Syn and measured gene expression in berries and cultured cells. Contrary to our hypothesis, we found that Syn increases anthocyanin accumulation via the production of hydrogen peroxide (H_2_O_2_), not phytohormones. We propose a mechanism underlying Syn-mediated grape skin coloration and discuss the coloration effect of H_2_O_2_.

## 2. Results

### 2.1. Syn Increases Anthocyanin Accumulation in VR Cells

We used VR (*Vitis* Red) cells to investigate the effect of Syn on anthocyanin accumulation (Figure 1). Anthocyanin content was significantly higher in VR cells treated with Syn concentrations of 1 mM or higher than in the control (*n* = 3, Tukey, *p* < 0.01 or 0.05; Figure 1a). We found no significant difference among VR cells treated with Syn concentrations of 1 mM or higher.

Syn is biosynthesized through the phenylalanine (Phe) and tyrosine (Tyro) pathways via L-(-)-tyrosine (L-Tyro), tyramine (Tyra), and octopamine hydrochloride (Oct) using Phe as the substrate [27]. Anthocyanin accumulation tended to decrease in the order of Phe, Syn, Oct, Tyra, and L-Tyro treatments. Anthocyanin contents were significantly higher in Syn- and Phe-treated VR cells than in the control (*n* = 3, Dunnett, *p* < 0.05 or 0.01; Figure 1b). The results indicate that Syn increased anthocyanin accumulation in VR cells in a concentration-independent and molecular-structure-specific manner.

### 2.2. Syn Increases Anthocyanin Accumulation in Grape Skin in Field Trials

We investigated whether Syn promotes grape ripening in the field by conducting field trials in 2019 and 2021 (Figure 2a). In 2019, anthocyanin content was significantly higher in Syn-treated berries than in the control on days 10 and 20 after treatment (*n* = 3, *t*-test, *p* < 0.01 or 0.05; Figure 2b). Similarly, in 2021, anthocyanin content was significantly higher in Syn-treated berries than in the control on day 20 after treatment (*n* = 3, *t*-test, *p* < 0.01; Figure 2c). However, the sugar/acid ratio, a ripeness index, was not significantly different between Syn-treated berries and the control, even though the scale on the y-axis differed between the two years (Figure 2d,e). The results indicate that Syn increased anthocyanin accumulation in grape skin but not berry ripening.

### 2.3. Syn Increases Anthocyanin-Biosynthesis-Related Gene Expression

We measured the expression of genes in anthocyanin-biosynthesis-related pathways (Figure 3). In the phenylpropanoid biosynthetic pathway, the relative expression of *PAL* encoding phenylalanine ammonia-lyase [EC 4.3.1.24] and *4CL* encoding 4-coumarate-CoA ligase [EC 6.2.1.12] was significantly higher (*n* = 4, *t*-test, *p* < 0.01 or 0.05) in Syn-treated VR cells than in the control at 24 h after treatment or later. In contrast, the relative expression of *C4H* encoding cinnamate-4-hydroxylase [EC 1.14.14.91] was significantly higher in Syn-treated VR cells than in the control at 72 h after treatment or later (*n* = 4, *t*-test, *p* < 0.01 or 0.05).

Upstream of the flavonoid biosynthetic pathway, the relative expression of *CHS* encoding chalcone synthase [EC 2.3.1.74] in Syn-treated VR cells was significantly different from that in the control only at 72 h after treatment, whereas the relative expression of *CHI* encoding chalcone isomerase [EC 5.5.1.6] showed a significant difference as early as 24 h after treatment (*n* = 4, *t*-test, *p* < 0.01). Midstream of the flavonoid biosynthetic pathway, the relative expression of *F3’H* encoding flavonoid 3′-monooxygenase [EC 1.14.14.82] and *F3’5’H* encoding flavonoid 3′,5′-hydroxylase [EC 1.14.14.81], which are related to red and blue anthocyanin pigment biosynthesis, differed in Syn-treated VR cells; *F3’H* showed a significant difference from the control at 48 h after treatment or later, whereas *F3’5’H* showed a significant difference at 96 h or later (*n* = 4, *t*-test, *p* < 0.01 or 0.05). The relative expression of *F3H* encoding flavanone 3-hydroxylase [EC 1.14.11.9] in Syn-treated VR cells was significantly different from that in the control at 48 h after treatment or later (*n* = 4, *t*-test, *p* < 0.01 or 0.05). Downstream, the relative expression of *DFR* encoding dihydroflavonol 4-reductase [EC 1.1.1.219] and *LDOX* encoding leucoanthocyanidin dioxygenase [EC 1.14.20.4] in Syn-treated cells was significantly different from those in the control at 72 h after treatment or later (*n* = 4, *t*-test, *p* < 0.01). In the flavonoid biosynthetic pathway, significant differences in the relative expression levels of these genes were observed in the early stages of the pathway.

The relative expression of *UFGT* encoding UDP-glucose:anthocyanidin/flavonol 3-O-glucosyltransferase [EC 2.4.1.115], a key enzyme in the anthocyanin biosynthetic pathway [28], and its transcription factor *mybA1* encoding Myb-related transcription factor A1 [29] was analyzed. The relative expression of *mybA1* in Syn-treated VR cells was significantly higher than that in the control from 24 h after treatment, and that of *UFGT* from 48 h after treatment (*n* = 4, *t*-test, *p* < 0.01 or 0.05). Overall, the results demonstrate that Syn increased the expression of genes in the anthocyanin-biosynthesis-related pathways as early as 24 h after treatment.

### 2.4. Syn Does Not Increase the Production of Phytohormones That Promote Anthocyanin Accumulation

We measured the relative expression of *NCED1* encoding 9-cis-epoxycarotenoid dioxygenase [EC 1.13.11.51] and *ACS3* encoding 1-aminocyclopropane-1 carboxylate synthase [EC 4.4.1.14], the rate-limiting enzymes of ABA and ET, respectively, in VR cells. We found that Syn did not increase *NECD1* expression or ABA content at 24 h after treatment (Figure 4a,b). The relative expression of *ACS3* in Syn-treated VR cells was not significantly different from that in the control at 0 and 12 h after treatment but was significantly different at 24 h after treatment (*n* = 4, *t*-test, *p* < 0.05) (Figure 4c). Because of technical difficulties in the quantification of volatile gas ET, we measured the relative expression level of *ACO2* encoding aminocyclopropanecarboxylate oxidase [EC 1.14.17.4], a key enzyme in ET biosynthesis, and found that the expression was not significantly different between Syn-treated VR cells and the control from 0 to 24 h after treatment (Figure 4d). 

We found that the relative expression of *LOX* encoding linoleate 13S-lipoxygenase [EC 1.13.11.12], the rate-limiting enzyme in the JA biosynthetic pathway, was significantly different between Syn-treated VR cells and the control at 12 h after treatment (*n* = 4, *t*-test, *p* < 0.05, Figure 4e). We also investigated the effect of Syn on the biosynthesis of JA, a phytohormone that increases berry skin coloration and disease resistance. Endogenous JA content in Syn-treated VR cells was not significantly different from that in the control at 24 h after treatment (Figure 4f). The results indicate that Syn is not involved in the biosynthesis of phytohormones that increase skin coloration.

### 2.5. Syn Increases Anthocyanin Content via H_2_O_2_

The relative expression of *SOD3* encoding the H_2_O_2_-generating enzyme superoxide dismutase [EC 1.15.1.1] was significantly higher in Syn-treated VR cells than in the control as early as 3 h to 12 h after treatment. H_2_O_2_ content in the Syn-treated cells was significantly higher than that in the control from 3 h to 24 h (*n* = 4, *t*-test, *p* < 0.01 or 0.05; Figure 5a,b). We also measured the relative expression of *APX6* and *CAT* encoding H_2_O_2_-scavenging enzymes ascorbate peroxidase [EC 1.11.1.11] and catalase [EC 1.11.1.6], respectively, as H_2_O_2_-responsive genes. The relative expression of *APX6* was significantly higher (*n* = 4, *t*-test, *p* < 0.01 or 0.05) at 24 h, and that of *CAT* at 24 h and 48 h, after Syn treatment compared with the control (Figure 5c,d).

We measured the relative expression of H_2_O_2_-responsive gene *Chit4* encoding class 4 chitinase [EC 3.2.1.14] in VR cells and found that it was significantly higher (*n* = 4, Dunnett, *p* < 0.01) 4 days after the treatment with Syn and H_2_O_2_ than in the control (Figure 6a). Similarly, the relative expression of *mybA1* and *UFGT* in VR cells showed a significant increase 4 days after the treatment with Syn and H_2_O_2_, and anthocyanin content was significantly higher 7 days after the treatment (*n* = 4, Dunnett, *p* < 0.01 or 0.05) than in the control (Figure 6b–d). The anthocyanin content in berry skin increased 9 days after the treatment with Syn and H_2_O_2_ (*n* = 4, Dunnett, *p* < 0.01 or 0.05) (Figure 6e). The results show that Syn increased anthocyanin accumulation via H_2_O_2_.

## 3. Discussion

We propose a mechanism by which Syn increases anthocyanin accumulation via H_2_O_2_ production and not phytohormones to solve the problem of poor grape skin coloration due to global warming (Figure 7).

We showed that Syn increases *SOD3* at a very early stage of treatment and generates H_2_O_2_. The Syn analog β-phenylethylethylamine promotes the immediate and transient generation of H_2_O_2_ as a product of the phenylethylamine degradation reaction by monoamine oxidase (MAO) in yeast [30,31], tobacco [32,33], and mesenchymal stem cells [34]. These findings suggest that Syn is an early inducer of H_2_O_2_ in grape cells, similar to its analog β-phenylethylethylamine.

We revealed that Syn increases anthocyanin accumulation in grapes in the same manner as the treatment with H_2_O_2_. The accumulation of the antioxidant anthocyanin confers H_2_O_2_-mediated oxidative stress tolerance to plants [35]. Consistent with our results, H_2_O_2_ increases anthocyanin accumulation in many plant species including grapes [36,37,38,39,40]. In addition, anthocyanin from apple peel can remove H_2_O_2_ better than other phenolics [41]. On the other hand, *Chit4* expression can be considered a marker of H_2_O_2_-mediated oxidative stress response [42,43,44]. Our finding that Syn and H_2_O_2_ upregulated the expression of *Chit4* and anthocyanin-biosynthesis-related genes, which in turn increased anthocyanin accumulation in grapes, suggests that Syn increases the accumulation of the antioxidant anthocyanin by inducing H_2_O_2_-mediated oxidative stress in grape cells. However, the balance between H_2_O_2_ production and removal is strictly regulated because excess H_2_O_2_ causes oxidative injury to cells [45,46]. Therefore, we assume that balancing H_2_O_2_ production and oxidative stress by regulating Syn concentration is important for H_2_O_2_-mediated anthocyanin accumulation by Syn because excess H_2_O_2_ may lead to anthocyanin degradation.

We indirectly showed that Syn did not increase ABA, ET, and JA contents, nor did it increase the sugar/acid ratio, a ripeness index related to ABA and ET. Syn only increased anthocyanin accumulation in grape berries. On the basis of these findings, we would like to emphasize that Syn-derived H_2_O_2_ is a useful coloration factor independent of phytohormones. Previous studies have focused on phytohormones to improve grape skin coloration [4,6,9,17,18,19,20,21]. H_2_O_2_ is a signaling molecule that regulates physiological processes such as plant growth and stress response, and crosstalk exists between H_2_O_2_ and phytohormones [45,47,48]. This crosstalk requires further evaluation.

As one of the limitations of this study, we were unable to consider alternative pathways to MAO for the generation of H_2_O_2_ from Syn. This is because there are no reports of Syn analogs producing H_2_O_2_ directly or indirectly via SOD except for the results of this study. The quantification of superoxide and SOD activities is needed to clarify this. He et al. (2020) reported a Syn–HCl-mediated reduction of H_2_O_2_ levels in postharvest litchi, in contrast to our findings in grapes [49]. These differences in results may be attributed to differences in Syn concentration, species-specific metabolic pathways, and H_2_O_2_ mitigation mechanisms. Furthermore, the physiological state of the fruit before and after harvest may influence these results. Comprehensive comparative studies with different plant species and growth stages are needed to elucidate species-specific responses and verify the broad applicability of Syn.

Syn has the potential to improve grape skin coloration by increasing H_2_O_2_ production without undesirable side effects such as defoliation [20] caused by ABA agents. Although there is concern about it being a health hazard because of its structural similarity to the doping agent ephedrine, Syn can be safely used as a dietary supplement [50,51]. The usefulness and safety of Syn as a grape color-enhancing agent should be evaluated by further field trials and H_2_O_2_ residual analysis. Syn application is expected to contribute to viticulture and the wine industry.

## 4. Materials and Methods

### 4.1. In Vitro Trials

Cultured grape cells (VR cells, PRC00003) were provided by the RIKEN BioResource Center Research (RIKEN BRC) through the National BioResource Project of MEXT/AMED, Japan. The cell line was derived from *Vitis* hybrid cv. Bailey Alicante A, which has high anthocyanin-biosynthesizing ability [52]. Modified Linsmaier and Skoog (LS) medium (pH 6.1) containing 3% (*w*/*v*) sucrose, 0.05 mg/L 2,4-dichlorophenoxyacetic acid (2,4-D), and 0.2 g/L kinetin was used. The medium was autoclaved (1.06 kg cm^−2^) at 121°C for 15 min, gelled with 1.2% (*w*/*v*) agar, and poured into disposable sterile plastic Petri dishes. Only white VR cells without red coloration were subcultured every week under sterile conditions and grown in a dark incubator at 27 °C.

In the coloration experiments, because phytohormones 2,4-D and kinetin inhibit anthocyanin accumulation during maturation, 10 mL of modified LS medium without phytohormones was autoclaved and dispensed into 70 × 16.5-mm-diameter Petri dishes. One dish was inoculated with 3–4 VR cells (each approximately 5 mm in diameter) under sterile conditions and incubated for up to 7 days in an incubator at 27 °C, 54.2 μmol m^−2^ s^−1^/16 h/day. The final concentrations of the test solutions in the medium were as follows: Syn concentrations of 0.01, 0.1, 1, 5, and 10 mM; molecular structure specificity, 5 mM each of phenylalanine (Phe), L-(-)-tyrosine (L-Tyro), tyramine (Tyra), octopamine hydrochloride (Oct), Syn, and control (Cont) (Tokyo Chemical Industry, Tokyo, Japan); H_2_O_2_ test, 10 mM H_2_O_2_ (30% H_2_O_2_, Fujifilm Wako, Osaka, Japan). Stock solutions of reagents were prepared and sterilized by filtration using a sterile syringe (2.5 mL SS-02SZ, Terumo, Tokyo, Japan) and a sterile filter (Minisart^®^ 0.45 µm syringe filter, Sartorius, Göttingen, Germany). Each sterile solution was added to the autoclaved medium and adjusted to the above concentrations.

### 4.2. Field Trials

*Vitis vinifera* cv. Syrah grapevines in the experimental vineyard (2019, 2021) of the Institute of Enology and Viticulture and an affiliated farm (2022) of the University of Yamanashi (at 35 °N, 138 °E in Yamanashi, Japan) were used. The grapevines were approximately 30 years old and grown using the double-cordon-style training method. 

A solution of 1 mM Syn with 0.01% (*v*/*v*) Approach BI (Kao, Tokyo, Japan) was prepared. The grapevines were defoliated in the berry zone before veraison and sprayed with 500 mL of water (control) or Syn solution per grapevine at veraison (30 July 2019; 18 July 2021). Thereafter, grape bunches were sampled every 10 days. The bunches were photographed and stored at −80 °C until RNA analyses.

Solutions of 1 mM Syn and 300 mM H_2_O_2_ with 0.01% (*v*/*v*) Approach BI (Kao, Tokyo, Japan) were prepared (10 August 2022). Nine grape bunches were randomly selected from one grapevine. Three bunches each were sprayed with water (control), Syn, and H_2_O_2_. On days 0 and 9 after spraying, bunches were harvested and 10 berries per bunch were randomly collected to determine anthocyanin content.

### 4.3. Total RNA Isolation

VR cells, 300 μL of Fruit-mate^TM^ for RNA Purification (TaKaRa, Shiga, Japan), and 300 μL of Buffer RLT for use with an RNA extraction kit (RNeasy Plant Mini Kit, QIAGEN, Hilden, Germany) were added to a 2.0 mL tube. The mixture was homogenized (30.0 Hz, 3 min) using TissueLyser II (QIAGEN) and centrifuged at 120 rpm for 3 min at 4 °C. Total RNA was extracted from 450 μL of supernatant using QIAcube (QIAGEN) with the RNeasy Plant Mini Kit and accessories following the product manual.

### 4.4. Real-Time RT-PCR

Single-stranded cDNA was synthesized from total RNA using a PrimeScript^TM^ RT Reagent Kit with gDNA Eraser (TaKaRa) and TaKaRa Cycler Dice^TM^ mini (TaKaRa) following the manufacturer’s manual. Real-time RT-PCR was performed using TB Green Premix Ex Taq II (Tli RNaseH Plus) (TaKaRa) with Thermal Cycler Dice Real Time System III (TaKaRa) following the manufacturer’s manual. Data were analyzed using Thermal Cycler Dice^®^ Real-Time System Single Software ver. 5.11. Actin was used for normalization because it is recommended as a reference gene for grapes, and expression levels are shown as relative values [53]. The real-time RT-PCR conditions were as follows: 37 °C for 15 min for RT reaction and 85 °C for 5 s for cDNA synthesis, and then 40 cycles at 95 °C for 5 s and 60 °C for 30 s for PCR amplification. The nucleotide sequences of the primers were as follows: *Vvactin* (5′-CAAGAGCTGGAAACTGCAAAGA-3′ and 5′-AATGAGAGATGGCTGGAAGAGG-3′, GenBank accession no. AF369524), *PAL* (5′-AAACAAGGTGGTGCCCTTCA-3′ and 5′-GGTGTTGATCCTCACGAGCA-3′, NM_001397918), *C4H* (5′-AAAGGGTGGGCAGTTCAGTT-3′ and 5′-GGGGGGTGAAAGGAAGATAT-3′, XM_002266202), *4CL* (5′-AGATGGGGATCAAGCAAGGC-3′ and 5′-ATCTCGGCCGGCATGTAAAA-3′, XM_002272746), *CHS* (5′-TCTGAGCGAGTATGGGAACATG-3′ and 5′-CTGTGCTGGCTTTCCCTTCT-3′, NM_001280950), *CHI* (5′-GACGGGTCGCCAGTATTCAG-3′ and 5′-GCTTTGGCTTCTGCGTCAGT-3′, NM_001281104), *F3’H* (5′-TATGGGCTGACCCTACAACGA-3′ and 5′-CCTGGGCAAACAACCTCATT-3′, NM_001280987), *F3’5’H* (5′-AGGGTCGGAGTCAAATGAGTTC-3′ and 5′-CGCTGGATCCCTTGGATGT-3′, NM_001281235), *F3H* (5′-CCAATCATAGCAGACTGTCC-3′ and 5′-TCAGAGGATACACGGTTGCC-3′, NM_001281105), *DFR* (5′-AACTGCTCTTTCCCCGA-3′ and 5′-AACGTCCCTCTGCCTTAGGATTC-3′, NM_001281215), *LDOX* (5′-GCGATATGACCATCTGGCCTAA-3′ and 5′-ATCCCAACCCAAGCGATAGC-3′, NM_001281218.1), *mybA1* (5′-GCAAGCCTCAGGACAGAAGAA-3′ and 5′-ATCCCAGAAGCCCACATCAA-3′, AB111101), *UFGT* (5′-CTTCTTCAGCACCAGCCAATC-3′ and 5′-AGGCACACCGTCGGAGATAT-3′, NM_001397857.1), *NCED1* (5′-GAGACCCCAACTCTGGCAGG-3′ and 5′-AAGGTGCCGTGGAATCCATAG-3′, NM_001281270.1), *ACS3* (5′-CCACCCCATACTACCCAGGA-3′ and 5′-TTGAGGCTGCGTTTTTGAGC-3′, XM_003635528.3), *ACO2* (5′-CAAATGGACGCTGTGGAAAA-3′ and 5′-ATGGCGGAGGAAGAAGGTACT-3′, NM_001280942.1), *LOX* (5′-TGGGCTGAAGCTTTTGATAG-3′ and 5′-CTTGGGCTTGGGTAGTAGT-3′, FJ858257) [54], *SOD3* (5′-GGCGATTCATCTACGGTTGTC-3′ and 5′-CCTCCGCCGTTGAACTTG-3′, NM_001281206) [55], *APX6* (5′-GCCCACTCTCCCCATTCTC-3′ and 5′-TGGAGTTTTGGCGGGAAAT-3′, XM_002282641) [55], *CAT* (5′-GGAGGATGAAGCCATAAGAG-3′ and 5′-GGCTGCAAGGGCAAGATA-3′, XM_003631877) [56], and *Chit4* (5′-CAATCGGGTCCTTGTGATTC-3′ and 5′-CAAGGCACTGAGAAACGCT-3′, U97522).

### 4.5. Total Anthocyanin Content

Anthocyanins in berry skins or VR cells were extracted using the procedure of Yokotsuka et al. (1999) [57] with modifications. Briefly, 10 randomly selected berries per bunch were peeled and the skin was crushed with liquid nitrogen using a mortar and pestle. One gram of crushed skin (or weighed VR cells) was immersed in 10 mL (or 500 µL) of 1% HCl–methanol overnight in the dark. The mixture was centrifuged at 10,000 rpm for 5 min, and the supernatant was diluted with 1% HCl–methanol to bring it within the absorbance measurement range. After mixing, absorbance was measured at 520 nm using a spectrophotometer (ASV11D-S, AS ONE, Osaka, Japan). Total anthocyanin content (malvidin-3-*O*-glucoside equivalent) in skin and VR cells was calculated using a published formula [58].

### 4.6. Sugar/Acid Ratio

Ten berries per bunch were pressed to obtain grape juice. The juice was centrifuged at 10,000 rpm for 5 min. The sugar (Brix)/acid ratio of the supernatant was measured using a pocket refractometer (PAL-BX|ACID2, ATAGO, Tokyo, Japan) following the manufacturer’s instructions. Sugar content and acid content represent the percentage concentration of soluble solids and that of total acid in the juice, respectively (Brix (%) and acid (%)).

### 4.7. Phytohormone Contents

Each phytohormone was quantified by ELISA. JA content in VR cells was measured following the manual for plant JA using an ELISA kit (MyBioSource, San Diego, CA, USA), as reported by Tsai et al. (2019) [59]. Briefly, VR cells cultured for 24 h and PBS (100 μL of PBS/10 mg of tissue) were added to a 2 mL Eppendorf tube and homogenized (30.0 Hz, 3 min) using TissueLyser II (QIAGEN). Then, 50 μL of the supernatant was centrifuged in a tabletop centrifuge and dispensed into a 96-well plate. Within 15 min after the addition of Stop Solution in the kit, absorbance was measured at 450 nm using an absorbance microplate reader, and JA content was calculated by the calibration curve method. Similarly, ABA content in VR cells cultured for 24 h was measured using a Plant Hormone ABA ELISA kit (CUSABIO, Wuhan, China) as reported by Enoki et al. (2017) [10].

### 4.8. H_2_O_2_ Content

H_2_O_2_ content was determined using a Cell Meter^TM^ Intracellular Fluorimetric Hydrogen Peroxide Assay Kit *Green Fluorescence* (AAT Bioquest, Sunnyvale, CA, USA) following the method of Nie et al. (2020) [60] with modifications. Briefly, VR cells and Component C assay buffer (200 mg/mL) were added to a 2 mL Eppendorf tube and the mixture was homogenized (30.0 Hz, 3 min) using a TissueLyser II (QIAGEN). The homogenate was separated using a tabletop centrifuge, and 50 μL of the supernatant was used as a test sample. After the reaction solution was added following the manufacturer’s instructions, the mixtures were incubated at room temperature for 20 min, and fluorescence intensity was measured at Ex/Em = 485/538 nm using a fluorescence microplate reader. H_2_O_2_ content was calculated using the calibration curve method.

### 4.9. Statistical Analysis

Data are presented as means ± standard error (SE) of three or four independent biological replicates. Statistical analysis was performed using BellCurve for Excel software ver. 3.20. (Social Survey Research Information, Tokyo, Japan) with the Student’s *t*-test, Tukey test, or Dunnett test.

## 5. Conclusions

We proposed a molecular mechanism for the Syn-mediated anthocyanin accumulation in grapes. Anthocyanin accumulation was increased not by phytohormones but by hydrogen peroxide and the upregulation of anthocyanin-biosynthesis-related genes in Syn-treated cells. Syn increased the expression of chitinase-encoding gene, one of stress response markers. The results suggest that Syn increases antioxidant anthocyanin accumulation by inducing oxidative stress mediated by hydrogen peroxide. The application of Syn to grape berries may be an alternative to the conventional use of phytohormone-related agents for improving grape skin coloration.

## Figures and Tables

**Figure 1 ijms-25-05912-f001:**
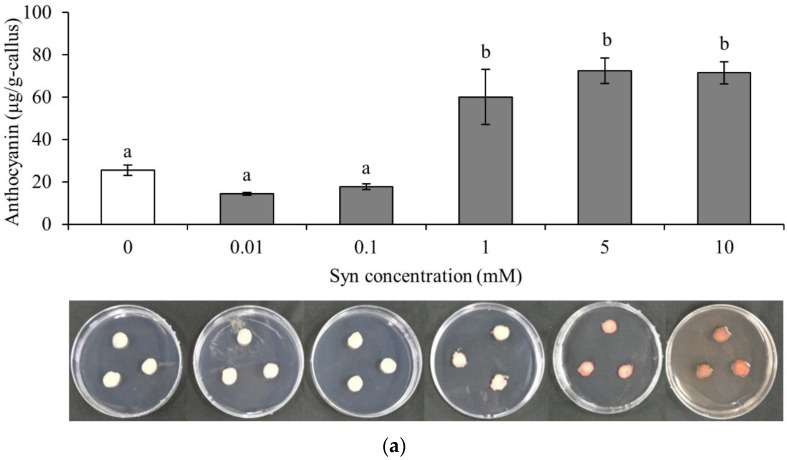
Effects of Syn on anthocyanin accumulation in VR cells. (**a**) Effects of Syn at concentrations of 0, 0.01, 0.1, 1, 5, and 10 mM on anthocyanin accumulation. Different letters (a,b) above the bar graphs indicate statistically significant differences (Tukey, *p* < 0.01 or 0.05). (**b**) Effects of Phe, L-Tyro, Tyra, Oct, and Syn (each 5 mM) on anthocyanin accumulation. * and ** indicate significant differences at *p* < 0.05 and 0.01, respectively, relative to control (Dunnett). VR cells were cultured at 27 °C, 54.2 μmol m^−2^ s^−1^/16 h/day for 7 days. Data are shown as means ± S.E. for three biological replicates (*n* = 3).

**Figure 2 ijms-25-05912-f002:**
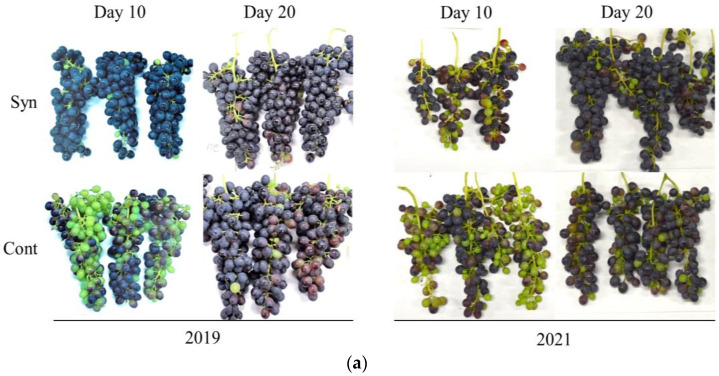
Effect of Syn on berry quality. Photographs of grape bunches 10 and 20 days after 1 mM Syn treatment in 2019 and 2021 (**a**). Anthocyanin content in berry skin (**b**,**c**). Sugar/acid ratio of juice (**d**,**e**). Data are shown as means ± SE (*n* = 3). * indicates significant difference at *p* < 0.05 and **, at 0.01 (*t*-test).

**Figure 3 ijms-25-05912-f003:**
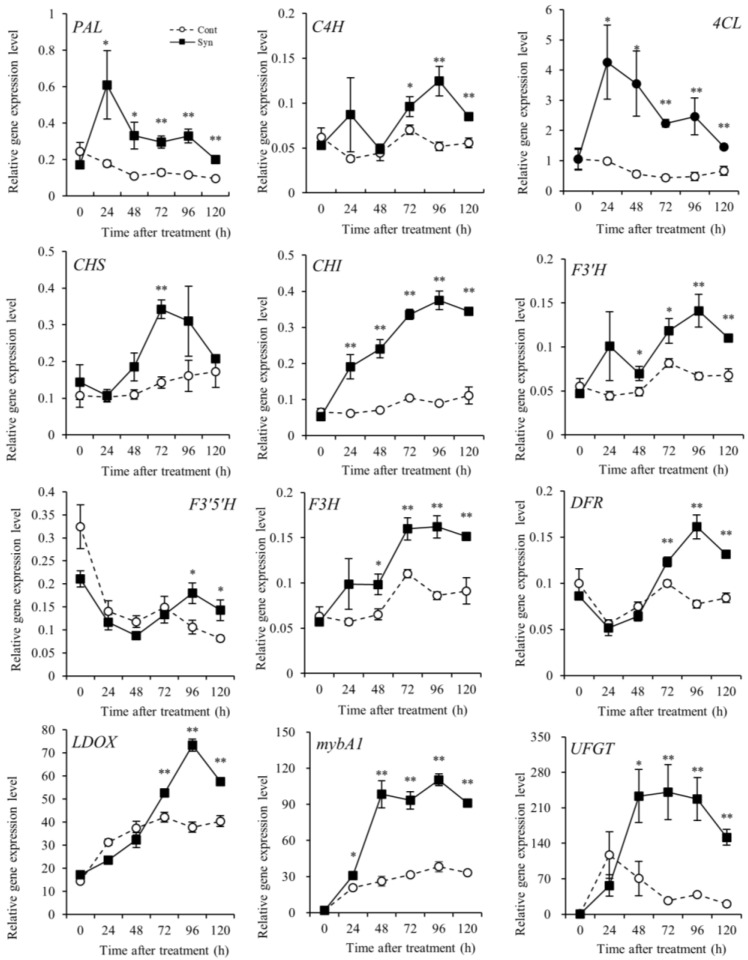
Expression levels of genes in anthocyanin-biosynthesis-related pathways in Syn-treated VR cells: *PAL*, *C4H*, and *4CL* in the phenylpropanoid biosynthetic pathway; *CHS*, *CHI*, *F3’H*, *F3’5’H*, *F3H*, *DFR*, and *LDOX* in the flavonoid biosynthetic pathway; and *mybA1* and *UFGT* in the anthocyanin biosynthetic pathway. VR cells were cultured in a medium containing 5 mM Syn for 120 h (27 °C, 54.2 μmol m^−2^ s^−1^/16 h/day). Gene expression level was estimated by real-time RT-PCR. Data are expression levels relative to actin and are shown as means ± S.E. of four biological replicates (*n* = 4). * and ** indicate significant differences at *p* < 0.05 and 0.01, respectively (*t*-test).

**Figure 4 ijms-25-05912-f004:**
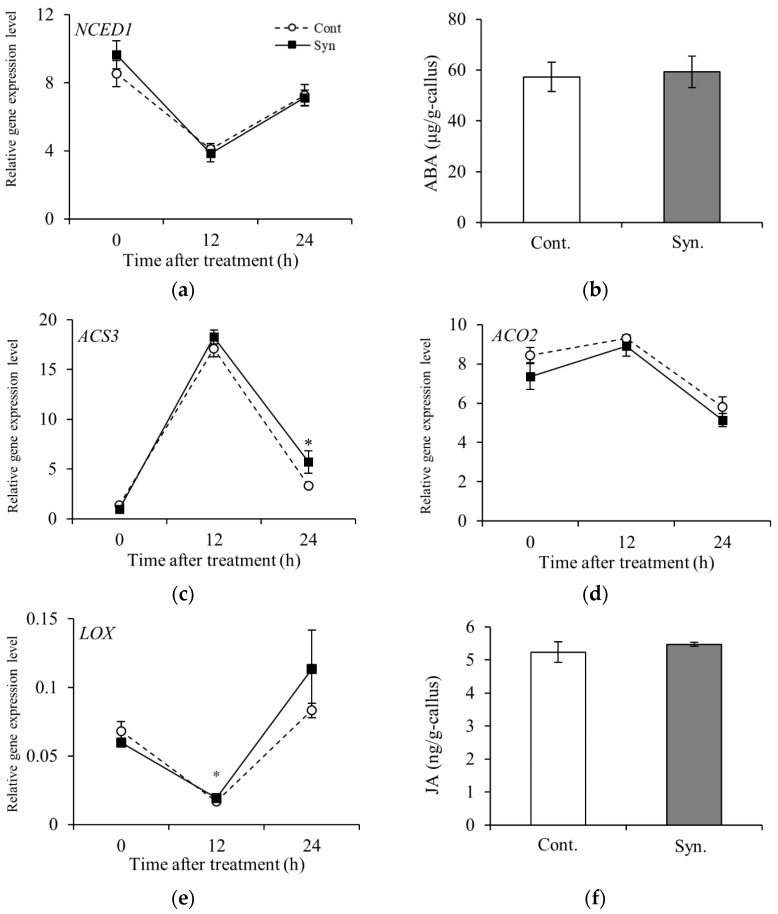
Expression levels of phytohormone biosynthesis genes and phytohormone contents in Syn-treated VR cells. Relative expression level of *NECD1* encoding ABA biosynthesis rate-limiting enzyme (**a**) and ABA content 24 h after treatment (**b**). Relative expression levels of *ACS3* (**c**) and *ACO2* (**d**), which encode the rate-limiting enzyme of the ET biosynthetic pathway and the enzyme that biosynthesizes ET, respectively. Relative expression level of *LOX* encoding JA biosynthesis rate-limiting enzyme (**e**) and JA content 24 h after treatment (**f**). VR cells were cultured for 24 h (27 °C, 54.2 μmol m^−2^ s^−1^/16 h/day) in a medium containing 5 mM Syn. Gene expression levels were estimated by real-time RT-PCR. Data are expression levels relative to actin. Phytohormone content was determined by ELISA. Data are shown as means ± S.E. of four biological replicates (*n* = 4). * indicates significant differences at *p* < 0.05 (*t*-test).

**Figure 5 ijms-25-05912-f005:**
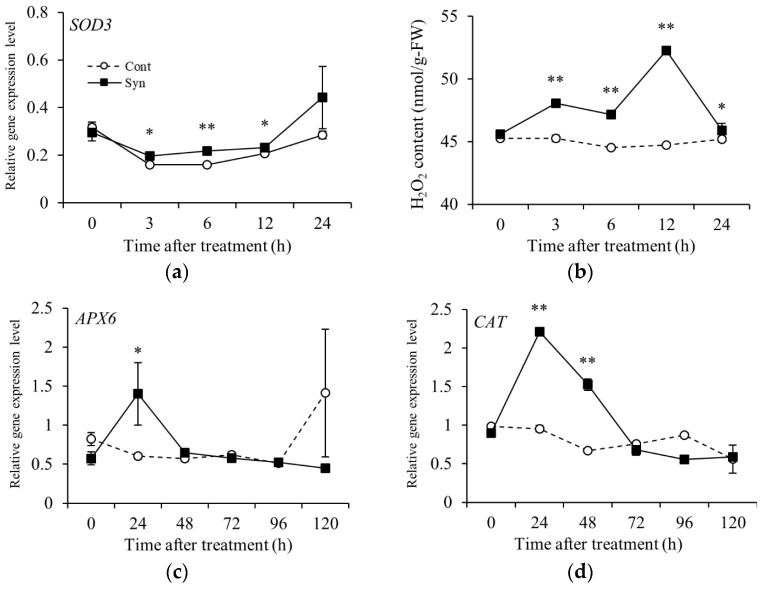
Relative expression levels of H_2_O_2_-related genes and H_2_O_2_ contents in Syn-treated cells. *SOD3* encoding H_2_O_2_-generating enzymes (**a**) and H_2_O_2_ content (**b**). Expression levels of *APX6* (**c**) and *CAT* (**d**) encoding H_2_O_2_-scavenging enzymes. VR cells were grown in a medium containing 5 mM Syn up to 24 or 120 h (27 °C, 54.2 μmol m^−2^ s^−1^/16 h/day). H_2_O_2_ content was measured with a fluorescence analysis kit. Gene expression levels were estimated by real-time RT-PCR. Data are expression levels relative to actin. Data are shown as means ± S.E. of four biological replicates (*n* = 4). * and ** indicate significant differences at *p* < 0.05 and 0.01, respectively (*t*-test).

**Figure 6 ijms-25-05912-f006:**
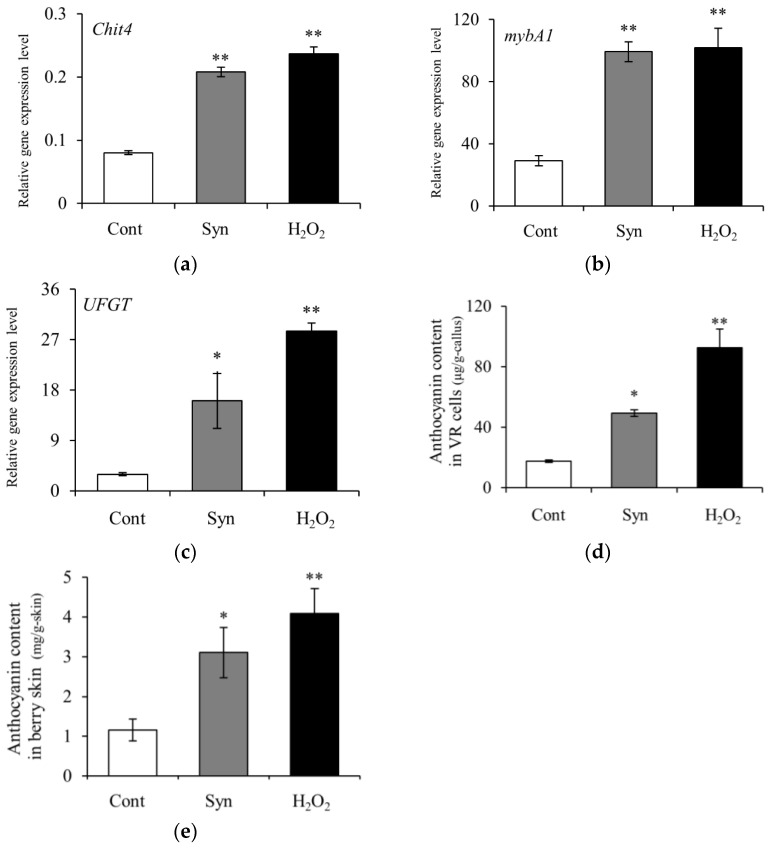
Effects of Syn and H_2_O_2_ treatments on VR cells and berries. Expression levels of H_2_O_2_-responsive gene *Chit4* (**a**) and anthocyanin-biosynthesis-related genes *mybA1* (**b**) and *UFGT* (**c**). Anthocyanin content in Syn- and H_2_O_2_-treated VR cells (**d**). Anthocyanin content in Syn- and H_2_O_2_-treated berry skin (**e**). VR cells were cultured in a medium containing 5 mM Syn and 10 mM H_2_O_2_ for 4 days (for gene expression levels) or 7 days (for anthocyanin content) at 27 °C, 54.2 μmol m^−2^ s^−1^/16 h/day. Berries were harvested 9 days after treatment with 1 mM Syn and 300 mM H_2_O_2_. Gene expression levels were estimated by real-time RT-PCR. Data are expression levels relative to actin. Data are means ± S.E. for four biological replicates (*n* = 4). * and ** indicate significant differences at *p* < 0.05 and 0.01, respectively (Dunnett test).

**Figure 7 ijms-25-05912-f007:**
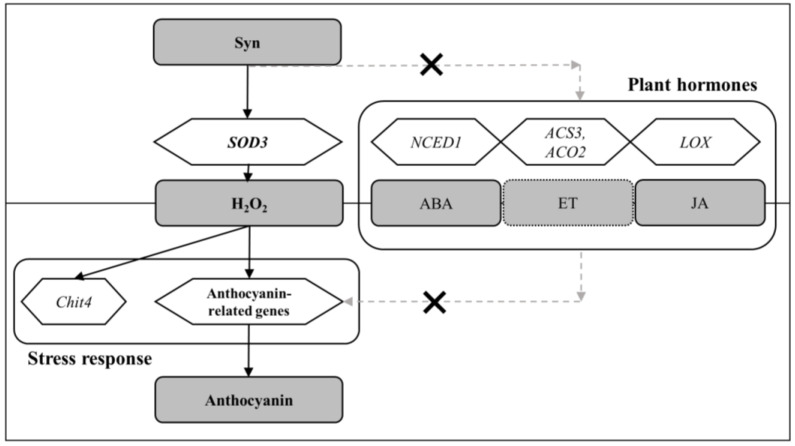
Proposed mechanism of Syn-mediated anthocyanin accumulation.

## Data Availability

The raw data supporting the conclusions of this article will be made available by the authors on request.

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
