# Peer review of "Application of Synephrine to Grape Increases Anthocyanin via Production of Hydrogen Peroxide, Not Phytohormones"

_ijms, 2024, doi:10.3390/ijms25115912_

Round 1
Reviewer 1 Report
Comments and Suggestions for Authors
The manuscript presents an interesting study on the potential use of synephrine to improve grape coloration and anthocyanin accumulation. However, there are several aspects that need to be critically analyzed for its suitability for publication.
The structure of the manuscript needs to be improved for better readability. It jumps between different sections without clear transitions, making it challenging to follow the logical flow of the study.
Even though statistical analyses are mentioned, specific details about sample sizes, replicates, and statistical tests performed are not provided in the Results section. Adding this information would enhance the rigor of the study.
The Discussion section should describe deeper into the effects of the findings and how they contribute to the existing knowledge in the field. Comparisons with previous studies or theories would strengthen the impact of the manuscript.
In conclusion, whereas the manuscript presents valuable research on synephrine’s potential in grape coloration enhancement, it requires significant revisions in terms of structure, language quality, and depth of analysis to meet the standards for publication.
Comments on the Quality of English Language
The quality of English language usage in the manuscript needs improvement. Some sentences are fragmented or unclear, affecting the overall readability.
Author Response
Answers to reviewers
Reviewer #1:
Comments and Suggestions for Authors
The manuscript presents an interesting study on the potential use of synephrine to improve grape coloration and anthocyanin accumulation. However, there are several aspects that need to be critically analyzed for its suitability for publication.
The structure of the manuscript needs to be improved for better readability. It jumps between different sections without clear transitions, making it challenging to follow the logical flow of the study.
Even though statistical analyses are mentioned, specific details about sample sizes, replicates, and statistical tests performed are not proved in the Results section. Adding thos information would enhance the rigor of the study.
The Discussion section should describe deeper into the effects of the findings and how they contribute to the existing knowledge in the field. Comparisons with previous studies or theories would strengthen the impact of the manuscript.
In Conclusion, whereas the manuscripts presents valuable research on synephrine’s potential in grape coloration enhancement, it requires significant revisions in terms of structure, language quality, and depth of analysis to meet the standards for publication.
[Answer]: Thank you for your valuable suggestions and comments.
・We have significantly improved the structure of the manuscript in the Introduction and Discussion sections. In the Discussion, we added comparisons with previous studies and highlighted the effects of our findings and their contributions to the existing knowledge in the field in the final paragraph.
(please see sections 1. Introduction and 3. Discussion)
・Regarding the analysis, we have included detailed information on sample sizes, replicates, and statistical tests performed in the Results section and figure legends, as well as in the statistical section of the Materials and Methods.
(please see 2. Results & Figure legend, L406-407)
Comments on the Quality of English Language
The quality of English language usage in the manuscript needs improvement. Some sentences are fragmented of unclear, affecting the overall readability.
[Answer]: Thank you for pointing this out. We reviewed the manuscript for language quality and have submitted a revised version for your consideration.

Reviewer 2 Report
Comments and Suggestions for Authors
Article
Application of synephrine to grape increases anthocyanin via production of hydrogen peroxide, not phytohormones
A brief summary
The themes of the works are aligned with contemporary trends and seek to provide solutions to the challenges facing the modern world. The title fit the work range and presented results. The study is really well designed, conducted, described and illustrated. The results are interesting and relevant. The conclusions concisely summarise the research.
Broad comments
1. The authors propose a solution to the poor skin colouration of the fruit and to enable the production of high-quality grapes of coloured varieties resulting from a temperature that may be too high during the fruiting period.
2. Particularly novel was the study of the effect of synephrine, a compound not previously used in plant cultivation, but used as a dietary supplement. By the way, it is very interesting to see how the authors came up with such an idea, especially as they do not cite any available literature that suggests such a possibility. This issue is somewhat enigmatic. An additional contribution of this research is that it proposes a possible mechanism of action for this compound that is supported by empirical evidence.
3. The data presented indicates that synephrine induces anthocyanin accumulation via the production of hydrogen peroxide and does not act as a phytohormone, which may has significant implications for the potential integration of this solution into horticultural practice.
4. It is of paramount importance to emphasise that a broad and multifaceted approach was employed to address the issue and also elucidate the underlying mechanisms governing fruit colouration.
5. Both the induction of anthocyanins in cells and in plants grown under field conditions were subjected to investigation. Interestingly, it has been shown in passing that synephrine only promotes anthocyanin accumulation in the grape skin and not fruit ripening. Synephrine has been demonstrated to enhance the expression of genes associated with anthocyanin biosynthesis. Importantly, it was shown that synephrine does not affect the increase in the production of phytohormones, which under standard conditions promote anthocyanin accumulation by examining its effect on speciffic gene expression. Finally, it has been demonstrated that synephrine increases the relative expression level of the gene encoding the enzyme that generates hydrogen peroxide, resulting in an increase in anthocyanin content. Both the description of the results and the discussion are presented in an appropriate manner and correctly illustrated. The conclusions provide a concise summary of the research findings.
6. The choice and development of literature is not objectionable. Although it may, however, resolve the verbal communication from line 53.
7. Additional comments and suggestions can be found below.
Specific comments
1. I have some reservations about the authors' choice of terminology, namely 'bioactive natural products'. This term is open to interpretation and may be perceived as marketing-oriented, particularly referring to the finished product offered by the retailer. In the context of this study, it would probably be more appropriate and secure to use the term ‘active compound’ or possibly ‘biologically active compound’.
2. I propose to change the key words. Some of them overlap with those used in the title, and there is also the phrase I mentioned a moment ago and the general statement 'phytohormone regulator'. I would rather suggest words such as: global warming, grape skin coloration, high-quality grape, sugar/acid ratio, SOD3, CAT, Chit4, mybA1, UFGT.
3. Line 4. The last author's name seems to be missing the last letter.
4. Line 149: Can the authors elaborate on the reason for the difficulty in ethylene determination?
5. Line 370: Please specify the units in which the instrument gives acid measurement results.
6. One could also have minor comments on the different scales in Figure 2 for each year.
Author Response
Answers to reviewers
Reviewer #2:
Open review
Does the introduction provide sufficient background and include all relevant references?
: Must be improved
[Answer] We will address your comments in detail in the Broad Comments section.
A brief summary
The themes of the works are aligned with contemporary trends and seek to provide solutions to the challenges facing the modern world. The title fit the work range and presented results. The study is really well designed, conducted, described and illustrated, the results are interesting and relevant. The conclusions concisely summaries the research.
Broad comments
- The authors propose a solution to the poor skin coloration of the fruit and to enable the production of high-quality grapes of colored varieties resulting from a temperature that may be too high during the fruit period.
- Particularly novel was the study of the effect of synephrine, a compound not previously used in plant cultivation, but used as a dietary supplement. By the way, it is very interesting to see how the authors came up with such as idea, especially as they do not cite any available literature that suggests such a possibility. This issue is somewhat enigmatic. An additional contribution of this research is that it proposes a possible mechanism of action for this compound that is supported by empirical evidence.
- The data presented indicates that synephrine induces anthocyanin accumulation via the production of peroxide and dose not act as a phytohormone, which may has significant implications for potential integration of this solution into horticultural practice.
- It is of paramount importance to emphasize that a broad and multifaceted approach was employed to address the issue and also elucidate the underlying mechanisms governing fruit coloration.
- Both the induction of anthocyanins in cells and in plants grown under field conditions were subjected to investigation. Interestingly, it has been shown in passing that synephrine only promotes anthocyanin accumulation in the grape skin and not fruit ripening. Synephrine has been demonstrated to enhance the expression of genes associated with anthocyanin biosynthesis. Importantly, it was shown that synephrine does not affect the increase in the production of phytohormones, which under standard conditions promote anthocyanin accumulation by examing its effect on specific gene expression. Finally, it has been demonstrated that synephrine increses the relative expression level of the encoding the enzyme that generates hydrogen peroxide, resulting in an increase in anthocyanin content. Both the description of the results and the discussion are presented in an appropriate manner and correctly illustrated. The conclutions provide a concise summary of the research findings.
- The choice and development of literature is not objectionable. Although it may. However, resolve the verbal communication from line 53.
- Additional comments and suggestions can be found below.
[Answer] Thank you for evaluating our manuscript with a brief summary and broad comments, and for providing your valuable suggestions.
Below, we respond to comments 2, 4, and 6.
2.Regarding the rationale of our study, we have explained in the second paragraph of the Introduction why we focused on "biologically active compounds." We further detailed how we identified synephrine (Syn) from these compounds (our laboratory screened many compounds using preliminary experimental approaches).
As there is no existing literature or studies suggesting such a possibility, our study aims to elucidate the coloring effect and mechanism of action of Syn in grapes.
(Please see the second paragraph of the Introduction.)
3.We have added this point to the third paragraph of the Introduction.
(Please see the third paragraph of the Introduction, L61-62.)
4.For the old line 53, we have used the term "personal communication" due to the reasons mentioned in point 2.
(Please see L47)
Specific comments
Thank you for your suggestions and comments. We respond as follows.
- I have some reservations about authors’ choice of terminology, namely ‘bioactive natural products’. This term is open to interpretation and may be perceived as marketing-oriented, particularly referring to the finished product offered by the retailer. In the context of this study, it would probably be more appropriate and secure to use the term ‘active compound’ of possibly ‘biologically active compound’.
[Answer]: We have revised the term "bioactive natural products" to "biologically active compounds" as you recommended.
(Please see L44, 46)
- I propose to change the key words. Some of them overlap with those used in the title, and there is also the phrase I mentioned a moment ago and the general statement ‘phytohormone regulator’. I would rather suggest words such as: global warming, grape skin coloration, high-quality grape, sugar/acid ratio, SOD3, CAT, Chit4, mybA1, UFGT.
[Answer]: We have revised the keywords as you recommended.
(Please see L25, 26)
- Line 4. The last author’s name seems to be missing the last letter.
[Answer]: We have corrected that.
- Line 149: Can the authors elaborate on the reason for the difficulty in ethylene determination? (L 153)
[Answer]: We have added an additional explanation regarding that. (Line 154, 155)
- Line 370: Please specify the units in which the instrument gives acid measurement results. (L379)
[Answer]: We added explanations for Acid and Sugar content as well. (Line 378, 379)
- One could also have minor comments on the different scales in Figure s for each year.
[Answer]: We have added descriptions of annual differences for the sugars and acids with large scale changes. (Line 94, 95)

Reviewer 3 Report
Comments and Suggestions for Authors
find my comments in the pdf attached

minor
Author Response
Answers to reviewers
Reviewer #3:
Thank you for your suggestions and derailed comments. We respond as follows.
- Commented [M1]: Provide list of abbreviations
[Answer]: We have added a list of abbreviations (L28-30).
- Commented [M2]: Please clarify whether duplicate or triplicate analyses were carried out for analysis.
[Answer]: We have added detailed clarification on this point in both the Results and Materials & Methods sections.
(please see 2. Results & Figure legend, L406-407)
- Commented [M3]: The paper's discussion is not appropriate and needs to be modified with some recent references and some excessive explanations removed. The authors must focus on discussing and comparing their findings with previous reports on this field. Overall, a comparison with existing literature is lacking and should be incorporated. A comprehensive discussion should include the results, the trend(s), the reasons for the trend(s) obtained, and a comparison with other studies.
[Answer]:
Thank you for your valuable feedback.
Regarding the discussion section, we have made significant revisions by removing excessive explanations and references, and restructuring the logic.
Overall, we have incorporated comparisons with existing literature into the discussion, particularly in the fifth paragraph.
(Pease see the Discussion for details.)
- Commented [M4]: Provide longitude and latitude.
[Answer]: We have noted them (L296).
- Commented [M5]: Why this temperature? Wouldn’t it affect the product quality and structure?
[Answer]:
We have provided additional clarification in L302. The temperature chosen is suitable for long-term preservation in genetic analysis and is a commonly used condition.
We believe there are no issues regarding the destruction of quality or structure of relevant fruit components or processed products, as there is no literature suggesting such concerns.
- Commented [M6]: Why didn’t you use HPLC for the anthocyanin profile? This is important.
[Answer]:
Thank you for your suggestion.
We acknowledge the importance of using HPLC for quantifying and profiling anthocyanin pigments. However, in the case of wine grapes, most of the anthocyanin composition is dominated by malvidin, and quantification based on total anthocyanins expressed as malvidin equivalents has been widely used in the literature for many years.
Indeed, in determining the ratio of red to blue hues, the expression ratio of F3'H/F3'5'H genes showed no dramatic increase in either gene expression, with both expression levels higher in the treatment group than the control. Therefore, it was considered that synephrine had a beneficial effect on the entire anthocyanin synthesis pathway and did not significantly influence changes in pigment composition.
Hence, in this study, total anthocyanin content was presented in the common form of malvidin due to cost and time constraints associated with measurement.
While our focus in this study was on elucidating the mechanism of total anthocyanin increase, we value your input for future horticultural applications utilizing this substance.
- Commented [M7]: What volume? The materials used and the details and conditions of experimental procedures have to be described with sufficient clarity, thus allowing qualified operators to repeat the work
- Commented [M8]: So no filtration?
[Answer]:
[M7] We have added specific volumes.
[M8] Yes, no filtration. The residue is completely removed by centrifugation under these conditions, then diluted, and the absorbance can be measured without problems (the purpose of dilution is also described in MM).
(please see L361-364)
- Commented [M9]: Please clarify whether duplicate or triplicate analyses were carried out for analysis.
[Answer]: Since it is a overlap, we mentioned it in our response to Commented [M2].
- Commented [M10]: The manuscript article requires grammar, sentence structure, and reference format revision. Overused Passive voice in the manuscript seems hard to read. Please try to reword the phrases in the active voice. Grammar and punctuation mistakes. For consistency, please use the manuscript in just one English style (a non-variant British or British style, American style, etc.). There are phrases with the verb in the wrong tense. Sentences with words misspelled. Words are overused or unnecessary.
[Answer]: Thank you for your comments.
We have undergone thorough English proofreading focusing on the areas you mentioned.
Additionally, we have received English proofreading for the references, including any additional citations.
(please see revised manuscript & References)

Round 2
Reviewer 1 Report
Comments and Suggestions for Authors
The issues raised in the feedback have been positively addressed in the revised manuscript. The brief description of changes made is given below:
The manuscript has been comprehensively revised to improve readability, strengthen the analysis, and deepen the impact of discussion. This includes incorporating clear transitions for better flow, adding statistical details to the Results section, and revising the Discussion to explore the significance of findings in relation to existing research. Moreover, the language quality has been improved to ensure clarity throughout the manuscript.
Reviewer 3 Report
Comments and Suggestions for Authors
accept
Comments on the Quality of English Languageminor